# Distribution and Characteristics of *Listeria* spp. in Pigs and Pork Production Chains in Germany

**DOI:** 10.3390/microorganisms10030512

**Published:** 2022-02-26

**Authors:** Verena Oswaldi, Stefanie Lüth, Janine Dzierzon, Diana Meemken, Stefan Schwarz, Andrea T. Feßler, Benjamin Félix, Susann Langforth

**Affiliations:** 1Institute of Food Safety and Food Hygiene, Department of Veterinary Medicine, Freie Universität Berlin, Königsweg 67, 14163 Berlin, Germany; janine.dzierzon@fu-berlin.de (J.D.); diana.meemken@fu-berlin.de (D.M.); susann.langforth@fu-berlin.de (S.L.); 2Animal Health Team, European Food Safety Authority (EFSA), Via Carlo Magno 1A, 43126 Parma, Italy; 3Veterinary Centre for Resistance Research (TZR), Freie Universität Berlin, Robert-von-Ostertag-Str. 8, 14163 Berlin, Germany; stefan.schwarz@fu-berlin.de (S.S.); andrea.fessler@fu-berlin.de (A.T.F.); 4Department Biological Safety, German Federal Institute for Risk Assessment (BfR), Diedersdorfer Weg1, 12277 Berlin, Germany; stefanie.lueth@bfr.bund.de; 5Institute of Microbiology and Epizootics, Centre of Infection Medicine, Department of Veterinary Medicine, Freie Universität Berlin, Robert-von-Ostertag-Str. 7, 14163 Berlin, Germany; 6Salmonella and Listeria Unit, Laboratory for Food Safety, French Agency for Food, Environmental and Occupational Health & Safety (ANSES), University of Paris-Est, 14, rue Pierre et Marie Curie, CEDEX, 94706 Maisons-Alfort, France; benjamin.felix@anses.fr

**Keywords:** *Listeria monocytogenes*, *Listeria innocua*, *Listeria welshimeri*, pork production, food safety, next-generation sequencing, MLST, SNP, antimicrobial resistance, food contamination

## Abstract

*Listeria* (*L*.) *monocytogenes* is a foodborne pathogen that can cause disease, mainly in elderly, pregnant or immunocompromised persons through consumption of contaminated food, including pork products. It is widespread in the environment and can also be found in asymptomatic carrier animals, for example, in different tissues of pigs. To learn more about their nature, 16 *Listeria* spp. isolates found in tonsils and intestinal content of pigs and 13 isolates from the slaughterhouse environment were characterized using next-generation sequencing (NGS). A wide distribution of clonal complexes was observed in pigs, as well as in the pork production chain, suggesting multiple sources of entry. Hypervirulent clones were found in pig tonsils, showing the potential risk of pigs as source of isolates causing human disease. The presence of closely related isolates along the production chain suggests a cross-contamination in the slaughterhouse or recontamination from the same source, strengthening the importance of efficient cleaning and disinfection procedures. The phenotypical antimicrobial resistance status of *L. monocytogenes* isolates was examined via broth microdilution and revealed a low resistance level. Nevertheless, genotypical resistance data suggested multiple resistances in some non-pathogenic *L. innocua* isolates from pig samples, which might pose a risk of spreading resistances to pathogenic species.

## 1. Introduction

*Listeria* (*L*.) *monocytogenes*, a Gram-positive bacterium, is the cause of human listeriosis, a rare foodborne illness with a high hospitalization and case-fatality rate [1]. Currently, 20 *Listeria* species are known [2], among which *L. monocytogenes* and *L. ivanovii* are considered mammalian pathogens. While infections caused by *L. ivanovii* are seldom and mainly affect ruminants, *L. monocytogenes* is associated with most human and animal listeriosis disease cases [3]. Other *Listeria* species are generally considered non-pathogenic, although there are some rare cases of disease reported caused by *L. innocua* [4,5] and *L. seeligeri* [6].

The species *L. monocytogenes* is subdivided into four evolutionary lineages [7,8,9,10] and 13 serovars. Almost all human disease cases are associated with one of three serovars: 4b, 1/2a or 1/2b [11]. The bacterium is further classified into sequence types (STs) via multilocus sequence typing (MLST) and into clonal complexes (CCs). STs are defined as the unique collective of alleles from seven housekeeping genes, and CCs are defined as clusters of STs sharing at least six alleles [12,13]. MLST is a reference method that allows for comparison of isolates and demonstrates that only a few frequent *L. monocytogenes* clones are globally distributed [14].

Its resilience against a wide range of environmental stressors, such as low nutrient availability, acidic conditions, high salinity, and a broad temperature range from 0 °C to 45 °C [11] give *L. monocytogenes* the ability to survive in different food sources. In addition, *L. monocytogenes* is widespread in the environment, requiring constant control of this pathogen in food production facilities [1]. It is also present in the intestines of asymptomatic animals and humans [15]. Regarding pigs, studies have confirmed the presence of the pathogen on carcasses and in different tissues [16], as well as in the farm environment and in feed [17]. The prevalence of *Listeria* spp. in pigs in German slaughterhouses was described by Oswaldi et al. [18]. *L. monocytogenes* is frequently found in pork products worldwide [19,20], especially in products intended to be eaten raw (ready-to-eat products); this poses a threat to public health [21]. Some studies have confirmed living pigs as the origin of *L. monocytogenes* found in pork [17,22]; others identified the slaughter and processing environment as the source of pork contamination [23,24]. Demaître et al. [25] reasoned that persistent isolates in the slaughterhouse environment are more likely the source of contamination than less common, presumably transient and sporadically introduced isolates, as routine cleaning and sanitizing have become ineffective. Next-generation sequencing (NGS) makes it possible to investigate the origin of isolates and provides information on virulence and resistance [26].

*L. monocytogenes* is a highly heterogeneous species in terms of pathogenicity and contains hypovirulent and hypervirulent clones, which are most likely to cause disease. Thus, knowledge about the clonal structure of isolates is needed to assess their potential to cause disease [27]. The major genes associated with virulence in *Listeria* are involved in the cell infectious cycle: *prfA*, *plcA*, *hly*, *mpl*, *actA* and *plcB* for pathogenicity island 1 and *inlA* and *inlB* for pathogenicity island 2 [28]. Isolates considered non-pathogenic do not possess these virulence genes [29].

Knowledge about the antimicrobial resistance (AMR) status of isolates that cause human disease is crucial for treatment and the prevention of fatal outcomes. *L. monocytogenes* is susceptible to a wide range of antimicrobial agents that are effective against Gram-positive bacteria, such as tetracyclines, erythromycin, ampicillin and gentamicin [30], and shows intrinsic resistance to fosfomycin and fusidic acid [31]. The multiple antimicrobial resistance of a *L. monocytogenes* isolate was first shown in 1988 [32], followed by various resistant strains from different sources, including food, as well as environmental and human clinical samples [33]. Poyart-Salmeron et al. [32] showed that resistance genes can be transferred between *Listeria* species and other bacteria by self-transmissible plasmids.

The aim of this study was to perform NGS on *Listeria* spp. isolates found in pigs and along the corresponding pork processing chain to gain knowledge about their MLST types, differences in SNPs, virulence genes and antimicrobial resistance genes. Moreover, we identified the presence of phenotypic resistance against antimicrobial agents in these *L. monocytogenes* via broth microdilution. These results may provide insights into the relatedness of *Listeria* spp. found in pigs and along the corresponding pork production chain, as well as their relevance for human disease.

## 2. Materials and Methods

### 2.1. Origin of Isolates

The isolates were recovered by sampling 430 fattening pigs and the slaughterhouse environment in two industrial high-capacity pig slaughterhouses in Germany, as described by Oswaldi et al. [18], later referred to as slaughterhouses A and B. Overall, 16 isolates of *Listeria* spp. originating from pigs and 13 isolates originating from the slaughterhouse environment were isolated. In slaughterhouse A, samples were collected from pig tonsils and intestinal content on four dates in the winter period of 2018/2019, later referred to as dates 1 to 4. On sampling dates 3 and 4, with two months in between, environmental samples were also taken from slaughter, cutting and processing environments during processing. These environmental samples were collected with sponges (Whirl-Pak™ Speci-Sponge™ Environmental Surface Sampling Bags, Nasco, Fort Atkinson, WI, USA) moisturized with 10 mL sterile 0.85% saline solution. The samples from pig tonsils in slaughterhouse B were taken in October 2019 (sampling dates 5 and 6); no environmental samples were taken due to organizational reasons. Sample transportation and examination, as well as confirmation of presumptive isolates, were performed as described by Oswaldi et al. [18]. All confirmed isolates were stored under cryopreservation conditions at −80 °C in cryovials (ROTI^®^Store cryovials, Carl Roth GmbH + Co. KG, Karlsruhe, Germany).

### 2.2. Sequencing and Bioinformatic Analysis of Sequences

Next-generation sequencing (NGS) was performed at the German Federal Institute for Risk Assessment (BfR) in Berlin, Germany. The DNA was isolated with a PureLink^TM^ Genomic DNA Mini Kit (Invitrogen^TM^, Carlsbad, CA, USA). For lysis, the PulseNet protocol for Gram-positive bacteria was used (https://www.cdc.gov/pulsenet/pdf/pnl32-miseq-nextera-xt.pdf; accessed on 22 April 2021). DNA concentration was quantified using a Qubit^TM^ dsDNA BR assay kit with a Qubit^TM^ 2.0 fluorometer (Invitrogen^TM^, Carlsbad, CA, USA). The sequencing library was prepared with an Illumina DNA prep kit (Illumina Inc., San Diego, CA, USA). Sequencing was performed in paired-end mode at 2 × 150 bp with an Illumina NextSeq 500 or at 2 × 300 bp with an Illumina MiSeq (Illumina Inc., San Diego, CA, USA). Trimming of sequencing raw reads and overall quality assessment was conducted using the pipeline AQUAMIS [34].

For single-nucleotide polymorphism (SNP) analysis, the pipeline snippySnake (https://gitlab.com/bfr_bioinformatics/snippySnake; accessed on 25 May 2021), based on Snippy (https://github.com/tseemann/snippy; accessed on 25 May 2021), was used. Three separate SNP analyses were carried out for the three different *Listeria* species found so as not to distort the analysis with interspecies variability. Optimal reference genomes for SNP analysis were automatically identified via the mash-based search included in the snippySnake workflow. Accordingly, NZ_CP026043.1 (strain FDAARGOS_58) was used as reference for the analysis of *L. mono cy togenes* strains, NC_003212.1 (strain Clip11262) was used as reference for the analysis of *L. innocua* strains, and NZ_LT906444.1 (strain NCTC11857) was used as reference for the analysis of *L. welshimeri* strains. Complete linkage clustering of SNP distance matrices was performed in R. Trees were exported using the *phylogram* package and visualized in iTOL [35]. Strains with single-digit SNP differences were rated as likely to be related to one another.

The BakCharak pipeline (https://gitlab.com/bfr_bioinformatics/bakcharak; accessed on 27 May 2021) was used for MLST determination (database: pubMLST), as well as for screening of sequences for antimicrobial resistance genes (database: NCBI resistance gene database) and virulence factors (database: VFDB).

### 2.3. Antimicrobial Susceptibility Testing

For all *L. monocytogenes* isolates, antimicrobial susceptibility testing was performed by broth microdilution according to the recommendations of the Clinical and Laboratory Standards Institute given in the document VET06 [36]. Bacterial suspensions with a turbidity equivalent to McFarland 0.5 were prepared. Subsequently, 5 µL of the suspension was mixed per mL CAMHB II (cation-adjusted Mueller-Hinton-II) broth (Oxoid, Wesel, Germany) supplemented with 5% (v/v) lysed horse blood. Then, 50 µL of this suspension was pipetted in each well of the four microtiter plates (Thermo Fisher Scientific, Basingstoke, UK) with a multichannel pipet and incubated at 35 °C ± 2 °C for 24 h. To count the colony-forming units (cfu), 100 µL of the suspension was inoculated on Columbia blood agar. *Streptococcus pneumoniae* ATCC^®^ 49619 was used as a quality control strain. The resulting minimal inhibitory concentrations (MICs) of penicillin, ampicillin and sulfamethoxazole/trimethoprim were classified as susceptible, intermediate or resistant according to the clinical breakpoints available in CLSI documents M45 [37] and VET06 [36]. For the remaining antimicrobial agents, erythromycin, clindamycin, ciprofloxacin, gentamicin, tetracycline and vancomycin, the clinical breakpoints for staphylococci listed in the CLSI document M100 [38] were used. Furthermore, based on CLSI document VET01S [39], breakpoints for staphylococci of animal origin were applied for amoxicillin/clavulanic acid, enrofloxacin, marbofloxacin, pirlimycin and doxycycline. For streptomycin and neomycin, the breakpoints described by Troxler et al. [31] were used.

## 3. Results

### 3.1. Presence of Listeria spp. in Pigs in Slaughterhouse A and B

Sixteen isolates of *Listeria* spp. (seven *L. monocytogenes* and nine *L. innocua*) used in this study were found in porcine samples; their distribution and prevalence in pigs were reported by Oswaldi et al. [18]. Two *L. monocytogenes* and five *L. innocua* isolates originated from slaughterhouse A, taken on sampling dates 1 to 4, whereas the remaining five *L. monocytogenes* and four *L. innocua* isolates came from slaughterhouse B on sampling dates 5 and 6.

### 3.2. Presence of Listeria spp. in Environmental Samples in Slaughterhouse A

The numbers of slaughter and processing environmental samples positive for *Listeria* spp. are shown in Table 1. 

No *Listeria* spp. was found in samples of saws (*n* = 8), tonsil removal devices (*n* = 4) or other equipment in the slaughter hall with product contact (*n* = 16). On date 3, *L. monocytogenes*, *L. innocua* and *L. welshimeri* were found on soles of rubber boots used within the slaughter hall. *L. monocytogenes* was isolated from the same drain on date 3 and date 4. On date 4, we found samples positive for *L. monocytogenes* in the cutting and processing plant in places with direct product contact, including the mincer. In the cutting plant, *L.*
*monocytogenes*-positive samples were found on the conveyor belt and a cutting tool. Other *Listeria* species, *L. innocua* and *L. welshimeri*, were found in the processing plant on both sampling dates.

### 3.3. MLST Analyses

The distribution of STs and CCs of the *L. monocytogenes* isolates is presented in Table 2. There were lineage I isolates with ST5 = CC5 *(n* = 2) and ST6 = CC6 (*n* =3), as well as lineage II isolates with ST7 = CC7 (*n* = 1), ST9 = CC9 (*n* =1), ST18 = CC18 (*n* =1), ST20 = CC20 (*n* = 1), ST37 = CC37 (*n* =1), ST325 = CC31 (*n* =1), ST412 = CC412 (*n* =2) and ST451 = CC11 (*n* = 1). Altogether, there were ten different STs and CCs present among the 14 isolates.

### 3.4. SNP Analyses

The results of the SNP analyses showed great genetic differences between the isolates, consistent with MLST types. Intra-CC diversity showed SNP differences, most likely related to the isolates’ epidemiological or evolutionary relationship.

The two *L. monocytogenes* isolates found in the same drain in the slaughter hall of slaughterhouse A on two different dates both belong to CC412 and showed a difference of 62 SNPs, which makes a direct relation unlikely (Figure 1). In the cutting plant, two *L.*
*monocytogenes* isolates found on a conveyor belt and a cutting tool further down the line belong to CC5 and had only one pairwise SNP difference, indicating a possible connection (Figure 1).

The two CC6 isolates found in pig tonsils in slaughterhouse B on the same date but originating from different farms have a pairwise SNP difference of 84, making a direct relation unlikely (Figure 1). In the processing plant of slaughterhouse A, another CC6 isolate was isolated from the meat mincer. The difference in SNPs was 43 to the nearer related CC6 isolate, so no close relationship was assumed. None of the isolated *L. innocua* showed close relationships with one another, as they all differed from each other by more than 90 SNPs (Figure 2).

Three clones of *L. welshimeri* isolated from the processing plant (mincer on dates 3 and 4, working surface in filling area on date 3) showed a close relationship to one another, with differences of only three and nine SNPs, respectively (Figure 3).

### 3.5. Antimicrobial Resistance Genes

All tested *Listeria* spp. isolates carried the AMR gene *fosX* (Table 2 and Table 3), which indicates genotypic resistance to fosfomycin.

In addition, all the *L. monocytogenes* isolates carried the gene *vga*(G), which confers resistance to lincosamides, streptogramin A antimicrobial agents and possibly pleuromutilins [40,41]. This gene has previously been referred to as *lmo0919* [40] and was later tentatively designated *vga*(L) [41]. The designation *vga*(G) was recently approved by the Nomenclature Center for Macrolide–Lincosamide–Streptogramin (MLS) Genes (https://faculty.washington.edu/marilynr/; accessed on 9 December 2021). As all *L. monocytogenes* share the same AMR genes, *fosX* and *vga*(G), and there was no difference in AMR gene content between *L. monocytogenes* of lineage I and II.

Among the 11 *L. innocua* isolates, seven harbored only the *fosX* resistance gene, whereas two isolates carried two, one isolate carried three and one isolate carried four AMR genes. One of these additional resistance genes was the gene *tet*(S), which codes for resistance to the tetracyclines doxycycline, tetracycline and minocycline. Three isolates carried the *tet*(M) gene, which confers the same resistance phenotype as *tet*(S). Two of these isolates additionally had the streptomycin resistance gene *ant*(*6*)*-Ia*, also known as *ant6* or *aadE*. One isolate harbored the trimethoprim resistance gene *dfrG* as a fourth resistance gene.

In one *L. welshimeri* isolate, the resistance gene *vga*(*G*) was found in addition to the *fosX* gene.

### 3.6. Virulence Genes

All *L. monocytogenes* isolates found in our study harbored the major virulence genes *prfA*, *plcA*, *hly*, *mpl*, *plcB*, *inlA* and *inlB*. Three isolates—two CC5s found in the cutting plant and one CC31 in a pig tonsil—carried a truncated version of *actA.* The truncation was of 1024/1920 nt and 1026/1920 nt for CC5 and CC31 strains, respectively, compared to EGD-e *act**A* (NCBI gene ID: 987035). The gene *actA* is one of the major virulence genes of pathogenicity island 1. Table 2 and Table 3 list the virulence genes for each isolate.

Neither *L. innocua* nor *L. welshimeri* isolates in this study showed any of the abovementioned major virulence genes.

### 3.7. Antimicrobial Susceptibility Testing Results

The results of antimicrobial susceptibility testing of 14 *L. monocytogenes* isolates are listed in Table 4. All tested isolates were susceptible to penicillin, ampicillin, erythromycin, ciprofloxacin, gentamicin, streptomycin, neomycin, tetracycline, sulfamethoxazole/trimethoprim and vancomycin. For amoxicillin/clavulanic acid, 57% (*n* = 8) of the *L.*
*monocytogenes* isolates tested susceptible, whereas 43% (*n* = 6) were classified as intermediate. All isolates, except of one intermediate isolate, were susceptible to the fluoroquinolone enrofloxacin according to the clinical breakpoints for staphylococci in VET01S. The majority (86%, *n* = 12) of the isolates tested susceptible for marbofloxacin, another fluoroquinolone, two isolates (14%) were classified as intermediate and 11 isolates (79%) had intermediate results recorded for doxycycline, whereas only 21% (*n* = 3) were susceptible. Using the clinical breakpoints for staphylococci in the CLSI documents M100 or VET01S, all tested isolates proved to be resistant to the lincosamides clindamycin and pirlimycin, which is in agreement with the carriage of the resistance gene *vga*(G).

## 4. Discussion

We were able to demonstrate a wide distribution of *Listeria* spp.-positive samples within the slaughterhouse and the associated meat production chain, with only five out of 29 *Listeria* spp. isolates being genetically related, including two *L. monocytogenes* and three *L. welshimeri* isolates, in contrast to 24 *Listeria* spp. isolates not being genetically related. This assumes several different sources of entry along the slaughter and processing line. Isolates of *L. monocytogenes* were found in pig tonsil samples, as well as in the slaughter, cutting and processing environment. *L. innocua* was isolated in samples of pig intestinal content, whereas isolates of *L. welshimeri*, in contrast, were only present in samples from the slaughter and processing environment.

In our study, equipment in the slaughter hall was not found to be a carrier of *Listeria* spp., whereas isolates were detected on the soles of rubber boots worn within the slaughter hall. Positive samples in floor drains support the risk of spreading clones within the slaughterhouse via boots or via air during washdown. Berrang and Frank [42] showed that contaminated floor drains in poultry processing plants can be the source of airborne spread of *Listeria* and can cross-contaminate food contact surfaces, equipment, and exposed products. Therefore, the authors recommended that workers act with caution so as to not spray hoses directly into drains.

Maury et al. [27] grouped prevalent CC types into food-associated (CC9 and CC121), infection-associated (CC1, CC2, CC4 and CC6) and intermediate clonal complexes (others), with various origins. In our study, infection-associated isolates (belonging to CC6) were found in two pig tonsil samples originating from separate farms. We found another CC6 isolate in the mincer, which constitutes a threat for the consumer if products get contaminated. These isolates have the same CC but differences in SNPs, which indicates no close relationship. Therefore, we were not able to detect a direct connection between isolates found in pigs and along the slaughter, cutting and processing chain. CC6 and CC5, which we found in the cutting environment, were considered by Félix et al. [24] to be ubiquitous and evenly spread CCs in the pork production chain. In another study, Maury et al. [43] suggested that hypovirulent isolates belonging to CC9 and CC121 are better adapted to the food processing environment, whereas hypervirulent isolates of CC6 are better adapted to the mammalian gut environment. CC9 and CC121 were both reported as the most frequent CCs in the pork processing environment, confirming their adaptation to the conditions of the production environment [24,44]. Demaître et al. [25] suggested the continuous introduction and repeated contamination with the most common CCs via incoming carcasses and/or that they persist in the concerned cutting plants, likely by genetic determinants contributing to their establishment. We found no CC121 isolate in our study and one CC9 isolate on a sole of a rubber boot. Félix et al. [24] connected CC9 to raw meat processing, from which isolates of this CC may have originated. On a rubber boot, isolates can be easily distributed throughout the whole slaughterhouse and possibly contaminate food. Our finding of a CC37 isolate in a pig tonsil is in accordance with the results of Félix et al. [24], who reported that this CC type is better adapted to pig farms than to the pork production environment. CC11 (ST451) is considered a hypervirulent clonal complex specific to central Europe, as it is found frequently there, mainly in dairy but also in meat products [45], and has also been reported as the cause of human listeriosis cases in Germany [46]. We found one CC11 isolate in a pig tonsil in slaughterhouse A. These findings highlight the potential risk of pigs as a source of pathogenic *L. monocytogenes* in food, as also other studies have suggested [47,48]. Looking at the SNP results, we found closely related *L. monocytogenes* distributed in the cutting plant, indicating a carryover along the processing line, from the conveyor belt to a cutting tool further down the cutting line.

In the processing plant, two *L. welshimeri* clones isolated on date 3 indicate a cross-contamination from a working surface to the mincer. On date 4, the same clone was again isolated from the same mincer. Despite daily disinfection, this clone showed a persistence over a period of two months, or it was reintroduced from the same source. Stoller et al. [49] showed a possible persistence of *L. monocytogenes* isolates belonging to CC9, CC121 and CC204 in meat production plants for at least four years, suggesting disinfectant failures and biofilm formation of the pathogen as possible causes of persistence. Even though *L. welshimeri* is considered non-pathogenic [50], it has similar growth characteristics and can thereby be considered a model of distribution for *L. monocytogenes* [28] and illustrate the risk of persistence and cross contamination in the slaughterhouse environment.

The virulence genes analyzed support the pathogenicity of 11 out of the 14 *L. monocytogenes* isolates found in this study. The three exceptions were due to the truncation of the *actA*, as recently described [51]. The *actA* gene plays a role in intracellular motility and intercellular spreading, a key determinant of *L. monocytogenes* virulence [52]. Domann et al. [52] demonstrated that strains without this gene are incapable of infecting adjacent cells and are distinctly less virulent. In our study, two CC5 isolates found in the cutting plant and a CC31 isolate found in a pig tonsil in slaughterhouse B harbored only a truncated rather than the full-length gene.

All *L. innocua* and *L. welshimeri* isolates found in this study can be considered non-pathogenic. In contrast, atypical, pathogenic *L. innocua* isolates able to cause human disease were shown to have the genes encoding for the pathogenicity island 1 and *inlA* [53]. A study in Brazil [54] found such isolates in the environment of pork processing plants.

The genotypic AMR results showed fosfomycin resistance in all *Listeria* spp. isolates, which is not remarkable, as intrinsic resistance is known [31]. In addition, in all *L. monocytogenes* isolates, a genotypic resistance to lincosamides was found, which has been shown to be a common native resistance in *L. monocytogenes* [33]. Four *L.*
*innocua* isolates in our study had a genetic resistance to tetracyclines, and two were also resistant to the aminoglycoside streptomycin. In addition, one isolate had a fourth resistance gene determining trimethoprim resistance. The *L. innocua* isolates with multiple resistance genes in our study all originated from samples from pigs. Tetracycline is one of the most often used antimicrobial agents in pig production in Germany, and aminoglycosides are also commonly used [55]; therefore, subinhibitory levels of these antimicrobials may be expected in the gastrointestinal tract and promote the occurrence of resistances [33]. Consequently, *L. innocua* isolates can be of concern, as reservoirs of these antimicrobial resistance genes can be transferred to *L. monocytogenes*, for example, in the gastrointestinal tract.

The MIC values of the *L. monocytogenes* isolates of this study provide information about the phenotypic situation of resistance against antimicrobial agents. However, our interpretation of these values was limited, as specific clinical breakpoints for *L. monocytogenes* do not exist for all antimicrobial agents. Therefore, we used existing clinical breakpoints for erythromycin, clindamycin, ciprofloxacin, gentamicin, tetracycline and vancomycin that are applicable to staphylococci, although not necessarily for staphylococci of porcine origin. As a consequence, the classifications obtained with these breakpoints have to be considered with caution. No *L. monocytogenes* isolate in our study showed a phenotypic resistance against antimicrobial agents commonly used for treatment of human listeriosis, including ampicillin or penicillin alone or in combination with gentamicin, whereas for patients allergic to β-lactams, the combination of trimethoprim and a sulfonamide is recommended. For treatment of listeriosis in pregnant woman, erythromycin is usually used, whereas bacteraemia is usually treated with vancomycin. Other antimicrobial agents, such as rifampicin, tetracycline, chloramphenicol and fluoroquinolones are used to a lesser extent [56]. A small percentage of isolates in our study were classified as intermediate, and no isolate was shown to be resistant. However, Alonso-Hernando et al. [56] reported increasing resistances to the fluoroquinolones enrofloxacin and ciprofloxacin and other antimicrobial agents, such as gentamicin, which is commonly used for treating human listeriosis. These emerging resistances, particularly multidrug resistances, represent a public health concern, as they may result in unsuccessful treatment of human disease cases [33].

## 5. Conclusions

In this study, we analyzed *Listeria* spp. isolates in the pork production chain, beginning from tonsils and intestinal content of fattening pigs, through the slaughter hall to the cutting and the processing plant in Germany. We found 14 *L. monocytogenes* isolates belonging to ten different clonal complexes. The isolates found in floor drains risk cross-contamination of pork products. Closely related isolates were identified in the cutting plant, suggesting contamination along the cutting line. Additionally, we found hypervirulent CC6 isolates of *L. monocytogenes*, known for causing a high risk of human listeriosis, in pig tonsils, which verifies pigs as a potential entry source into pork products. Closely related *L. welshimeri* isolates found on a working surface and in a nearby mincer on two different sampling dates indicate a risk of cross-contamination and persistence of *Listeria* spp. over a longer period. This highlights the importance of proper cleaning and disinfection procedures.

In our study, *L. monocytogenes* isolates were found to have a low level of antimicrobial resistance, as demonstrated by genotypical analyses and phenotypical antimicrobial resistance tests. No isolate of *L. monocytogenes* from pigs or the slaughterhouse environment showed resistance to commonly used antimicrobials for treatment of human listeriosis. However, we found non-pathogenic *L. innocua* had multiple resistance genes, which poses a risk for public health, as bacteria are able to pass on their resistance genes to related and unrelated species through horizontal transfer mechanisms. Therefore, monitoring of the resistance situation of non-pathogenic species is also required.

In conclusion, for the successful control and treatment of human listeriosis, an one health approach is needed, considering the origin of the food contamination and possible causes of resistances.

## Figures and Tables

**Figure 1 microorganisms-10-00512-f001:**
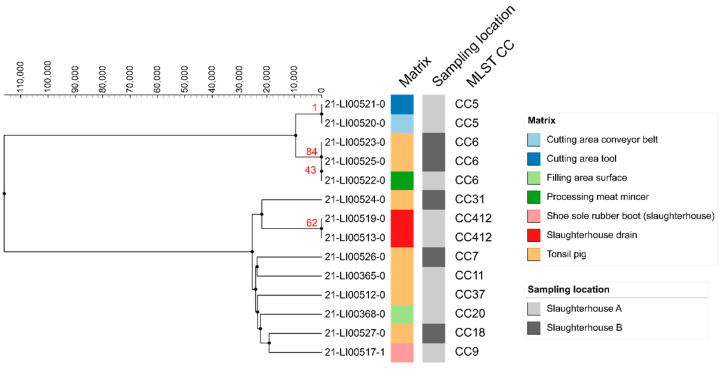
Clustering of 14 *L. monocytogenes* isolates found in pigs and the slaughterhouse environment in a linkage tree with differences of less than 1000 SNPs indicated.

**Figure 2 microorganisms-10-00512-f002:**
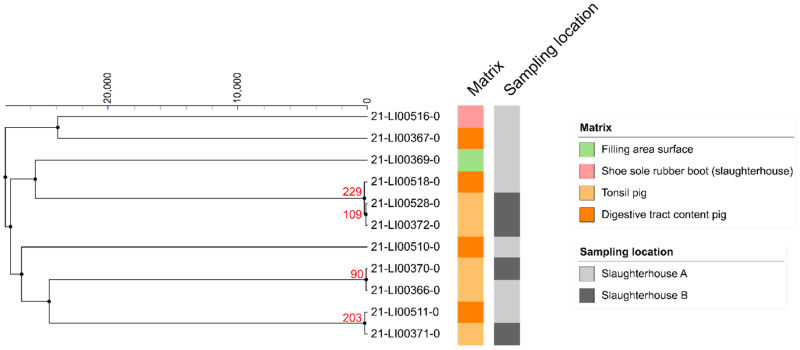
Clustering of 11 *L. innocua* isolates found in pigs and the slaughterhouse environment in a linkage tree with differences of less than 1000 SNPs indicated.

**Figure 3 microorganisms-10-00512-f003:**
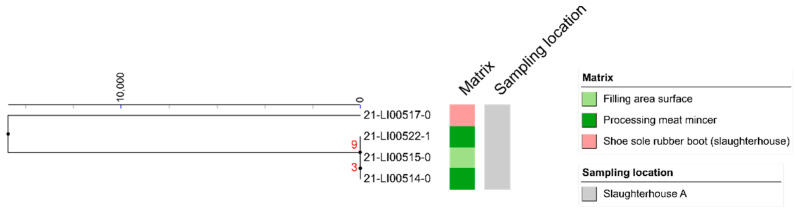
Clustering of 4 *L. welshimeri* isolates found in the slaughterhouse environment in a linkage tree with differences of less than 1000 SNPs indicated.

**Table 1 microorganisms-10-00512-t001:** Results of environmental samples taken on date 3 (*n* = 36) and date 4 (*n* = 41) in slaughterhouse A in absolute numbers.

Place of Sampling	Number of Samples Taken		Samples Positive for *L. monocytogenes*		Samples Positive for *L. innocua*		Samples Positive for *L. welshimeri*	
	Date 3	Date 4	Date 3	Date 4	Date 3	Date 4	Date 3	Date 4
Saws	4	4	0	0	0	0	0	0
Tonsil removal device	2	2	0	0	0	0	0	0
Floor drains	5	6	1	1	0	0	0	0
Rubber boots (sole)	7	8	1	0	1	0	1	0
Equipment in slaughter hall	8	8	0	0	0	0	0	0
Cutting plant (product contact surfaces/devices)	5	6	0	2	0	0	0	0
Processing plant (product contact surfaces/devices)	5	7	0	2	0	1	2	1
**Total**	**77**		**7**		**2**		**4**	

**Table 2 microorganisms-10-00512-t002:** Characteristics of *L. monocytogenes* isolates found in pigs and in the environment ^1^.

Sampled Matrix (Sample Number)	Slaughterhouse	Sampling Date	Lineage	MLST ST	MLST CC	Virulence Genes (Total Number)
Pig tonsil (21-LI00365-0)	A	2	II	451	11	*actA*, *ami*, *aut*, *bsh*, *clpC*, *clpE*, *clpP*, *fbpA*, *gtcA*, *hly*, *hpt*, *iap/cwhA*, *inlA*, *inlB*, *inlC*, *inlF*, *inlJ*, *inlK*, *lap*, *lapB*, *lntA*, *lpeA*, *lpA1*, *lspA*, *mpl*, *oatA*, *pdgA*, *plcA*, *plcB*, *prfA*, *prsA2*, *vip* (32)
Pig tonsil (21-LI00512-0)	A	3	II	37	37	*actA*, *ami*, *aut*, *bsh*, *clpC*, *clpE*, *clpP*, *fbpA*, *gtcA*, *hly*, *hpt*, *iap/cwhA*, *inlA*, *inlB*, *inlC*, *inlF*, *inlJ*, *inlK*, *lap*, *lapB*, *lntA*, *lpeA*, *lpA1*, *lspA*, *mpl*, *oatA*, *pdgA*, *plcA*, *plcB*, *prfA*, *prsA2* (31)
Pig tonsil (21-LI00523-0)	B	5	I	6	6	*actA*, *bsh*, *clpC*, *clpE*, *clpP*, *fbpA*, *gtcA*, *hly*, *hpt*, *iap/cwhA*, *inlA*, *inlB*, *inlC*, *inlF*, *inlK*, *lap*, *lapB*, *llsA*, *llsB*, *llsD*, *llsG*, *llsH*, *llsP*, *llsX*, *llsY*, *lntA*, *lpeA*, *lpA1*, *lspA*, *mpl*, *oatA*, *pdgA*, *plcA*, *plcB*, *prfA*, *prsA2*, *vip* (37)
Pig tonsil (21-LI00524-0)	B	5	II	325	31	*ami*, *aut*, *bsh*, *clpC*, *clpE*, *clpP*, *fbpA*, *gtcA*, *hly*, *hpt*, *iap/cwhA*, *inlA*, *inlB*, *inlC*, *inlF*, *inlJ*, *inlK*, *lap*, *lntA*, *lpeA*, *lpA1*, *lspA*, *mpl*, *oatA*, *pdgA*, *plcA*, *plcB*, *prfA*, *prsA2* (29)
Pig tonsil (21-LI00525-0)	B	5	I	6	6	*actA*, *bsh*, *clpC*, *clpE*, *clpP*, *fbpA*, *gtcA*, *hly*, *hpt*, *iap/cwhA*, *inlA*, *inlB*, *inlC*, *inlF*, *inlK*, *lap*, *lapB*, *llsA*, *llsB*, *llsD*, *llsG*, *llsH*, *llsP*, *llsX*, *llsY*, *lntA*, *lpeA*, *lpA1*, *lspA*, *mpl*, *oatA*, *pdgA*, *plcA*, *plcB*, *prfA*, *prsA2*, *vip* (37)
Pig tonsil (21-LI00526-0)	B	5	II	7	7	*actA*, *ami*, *aut*, *bsh*, *clpC*, *clpE*, *clpP*, *fbpA*, *gtcA*, *hly*, *hpt*, *iap/cwhA*, *inlA*, *inlB*, *inlC*, *inlF*, *inlJ*, *inlK*, *lap*, *lapB*, *lntA*, *lpeA*, *lpA1*, *lspA*, *mpl*, *oatA*, *pdgA*, *plcA*, *plcB*, *prfA*, *prsA2* (31)
Pig tonsil (21-LI00527-0)	B	5	II	18	18	*actA*, *ami*, *aut*, *bsh*, *clpC*, *clpE*, *clpP*, *fbpA*, *gtcA*, *hly*, *hpt*, *iap/cwhA*, *inlA*, *inlB*, *inlC*, *inlF*, *inlJ*, *inlK*, *lap*, *lapB*, *lntA*, *lpeA*, *lpA1*, *lspA*, *mpl*, *oatA*, *pdgA*, *plcA*, *plcB*, *prfA*, *prsA2*, *vip* (32)
Floor drain (21-LI00513-0)	A	3	II	412	412	*actA*, *ami*, *aut*, *bsh*, *clpC*, *clpE*, *clpP*, *fbpA*, *gtcA*, *hly*, *hpt*, *iap/cwhA*, *inlA*, *inlB*, *inlC*, *inlF*, *inlJ*, *inlK*, *lap*, *lntA*, *lpeA*, *lpA1*, *lspA*, *mpl*, *oatA*, *pdgA*, *plcA*, *plcB*, *prfA*, *prsA2* (30)
Sole of rubber boot (21-LI00517-1)	A	3	II	9	9	*actA*, *ami*, *aut*, *bsh*, *clpC*, *clpE*, *clpP*, *fbpA*, *gtcA*, *hly*, *hpt*, *iap/cwhA*, *inlA*, *inlB*, *inlC*, *inlF*, *inlJ*, *inlK*, *lap*, *lapB*, *lntA*, *lpeA*, *lpA1*, *lspA*, *mpl*, *oatA*, *pdgA*, *plcA*, *plcB*, *prfA*, *prsA2*, *vip* (32)
Floor drain (21-LI00519-0)	A	4	II	412	412	*actA*, *ami*, *aut*, *bsh*, *clpC*, *clpE*, *clpP*, *fbpA*, *gtcA*, *hly*, *hpt*, *iap/cwhA*, *inlA*, *inlB*, *inlC*, *inlF*, *inlJ*, *inlK*, *lap*, *lntA*, *lpeA*, *lpA1*, *lspA*, *mpl*, *oatA*, *pdgA*, *plcA*, *plcB*, *prfA*, *prsA2* (30)
Cutting plant (21-LI00520-0)	A	4	I	5	5	*ami*, *aut*, *bsh*, *clpC*, *clpE*, *clpP*, *fbpA*, *gtcA*, *hly*, *hpt*, *iap/cwhA*, *inlA*, *inlB*, *inlC*, *inlF*, *inlK*, *lap*, *lapB*, *lntA*, *lpeA*, *lpA1*, *lspA*, *mpl*, *oatA*, *pdgA*, *plcA*, *plcB*, *prfA*, *prsA2*, *vip* (30)
Cutting plant (21-LI00521-0)	A	4	I	5	5	*ami*, *aut*, *bsh*, *clpC*, *clpE*, *clpP*, *fbpA*, *gtcA*, *hly*, *hpt*, *iap/cwhA*, *inlA*, *inlB*, *inlC*, *inlF*, *inlK*, *lap*, *lapB*, *lntA*, *lpeA*, *lpA1*, *lspA*, *mpl*, *oatA*, *pdgA*, *plcA*, *plcB*, *prfA*, *prsA2*, *vip* (30)
Processing plant (21-LI00522-0)	A	4	I	6	6	*actA*, *bsh*, *clpC*, *clpE*, *clpP*, *fbpA*, *gtcA*, *hly*, *hpt*, *iap/cwhA*, *inlA*, *inlB*, *inlC*, *inlF*, *inlK*, *lap*, *lapB*, *llsA*, *llsB*, *llsD*, *llsG*, *llsH*, *llsP*, *llsX*, *llsY*, *lntA*, *lpeA*, *lpA1*, *lspA*, *mpl*, *oatA*, *pdgA*, *plcA*, *plcB*, *prfA*, *prsA2*, *vip* (37)
Processing plant (21-LI00368-0)	A	4	II	20	20	*actA*, *ami*, *aut*, *bsh*, *clpC*, *clpE*, *clpP*, *fbpA*, *gtcA*, *hly*, *hpt*, *iap/cwhA*, *inlA*, *inlB*, *inlC*, *inlF*, *inlJ*, *inlK*, *lap*, *lapB*, *lntA*, *lpeA*, *lpA1*, *lspA*, *mpl*, *oatA*, *pdgA*, *plcA*, *plcB*, *prfA*, *prsA2*, *vip* (32)

^1^ All the listed isolates showed AMR genes *fosX* and *vga*(G), as well as phenotypical resistance to clindamycin and pirlimycin.

**Table 3 microorganisms-10-00512-t003:** Characteristics of *Listeria* species other than *L. monocytogenes* found in pigs and the slaughterhouse environment.

Sampled Matrix (Laboratory Number)	Slaughterhouse	Sampling Date	Species	AMR Genes	Virulence Genes (Total Number)
Pig intestinal content (21-LI00510-0)	A	1	*L. innocua*	*fosX*; *tet*(S)	*clpC*, *clpE*, *clpP*, *fbpA*, *gtcA*, *iap/cwhA*, *lap*, *lpeA*, *lpA1*, *lspA*, *oatA*, *pdgA*, *prsA2* (13)
Pig intestinal content (21-LI00511-0)	A	2	*L. innocua*	*fosX*	*clpC*, *clpE*, *clpP*, *fbpA*, *gtcA*, *iap/cwhA*, *lap*, *llsA*, *llsG*, *llsH*, *llsX*, *lpeA*, *lpA1*, *lspA*, *oatA*, *pdgA*, *prsA2* (17)
Pig intestinal content (21-LI00518-0)	A	4	*L. innocua*	*fosX*; *tet*(M)	*clpC*, *clpE*, *clpP*, *fbpA*, *gtcA*, *iap/cwhA*, *lap*, *lpeA*, *lpA1*, *lspA*, *oatA*, *pdgA*, *prsA2* (13)
Pig intestinal content (21-LI00367-0)	A	4	*L. innocua*	*fosX*	*clpC*, *clpE*, *clpP*, *fbpA*, *gtcA*, *iap/cwhA*, *lap*, *llsA*, *llsG*, *llsH*, *llsX*, *lpeA*, *lpA1*, *lspA*, *oatA*, *pdgA*, *prsA2* (17)
Pig tonsil (21-LI00366-0)	A	2	*L. innocua*	*fosX*	*clpC*, *clpE*, *clpP*, *fbpA*, *gtcA*, *iap/cwhA*, *lap*, *lpeA*, *lpA1*, *lspA*, *oatA*, *pdgA*, *prsA2* (13)
Pig tonsil (21-LI00528-0)	B	6	*L. innocua*	*fosX*; *dfrG*; *tet*(M); *ant(6)-Ia*	*clpC*, *clpE*, *clpP*, *fbpA*, *gtcA*, *iap/cwhA*, *lap*, *lpeA*, *lpA1*, *lspA*, *oatA*, *pdgA*, *prsA2* (13)
Pig tonsil (21-LI00370-0)	B	6	*L. innocua*	*fosX*	*clpC*, *clpE*, *clpP*, *fbpA*, *gtcA*, *iap/cwhA*, *lap*, *lpeA*, *lpA1*, *lspA*, *oatA*, *pdgA*, *prsA2* (13)
Pig tonsil (21-LI00371-0)	B	6	*L. innocua*	*fosX*	*clpC*, *clpE*, *clpP*, *fbpA*, *gtcA*, *iap/cwhA*, *lap*, *llsA*, *llsG*, *llsH*, *llsX*, *lpeA*, *lpA1*, *lspA*, *oatA*, *pdgA*, *prsA2* (17)
Pig tonsil (21-LI00372-0)	B	6	*L. innocua*	*fosX*; *tet*(M); *ant(6)-Ia*	*clpC*, *clpE*, *clpP*, *fbpA*, *gtcA*, *iap/cwhA*, *lap*, *lpeA*, *lpA1*, *lspA*, *oatA*, *pdgA*, *prsA2* (13)
Processing plant (21-LI00514-0)	A	3	*L. welshimeri*	*fosX*	*clpC*, *clpE*, *clpP*, *fbpA*, *lap*, *lpeA*, *lpA1*, *lspA*, *prsA2* (9)
Processing plant (21-LI00515-0)	A	3	*L. welshimeri*	*fosX*	*clpC*, *clpE*, *clpP*, *fbpA*, *lap*, *lpeA*, *lpA1*, *lspA*, *prsA2* (9)
Sole of rubber boot (21-LI00516-0)	A	3	*L. innocua*	*fosX*	*clpC*, *clpE*, *clpP*, *fbpA*, *gtcA*, *iap/cwhA*, *lap*, *llsA*, *llsG*, *llsH*, *lpeA*, *lpA1*, *lspA*, *oatA*, *pdgA*, *prsA2* (16)
Sole of rubber boot (21-LI00517-0)	A	3	*L. welshimeri*	*fosX*	*clpC*, *clpE*, *clpP*, *fbpA*, *lap*, *lpeA*, *lpA1*, *lspA*, *prsA2* (9)
Processing plant (21-LI00522-1)	A	4	*L. welshimeri*	*fosX*; *vga*(G)	*clpC*, *clpE*, *clpP*, *fbpA*, *lap*, *lpeA*, *lpA1*, *lspA*, *prsA2* (9)
Processing plant (21-LI00369-0)	A	4	*L. innocua*	*fosX*	*clpC*, *clpE*, *clpP*, *fbpA*, *gtcA*, *iap/cwhA*, *lap*, *llsA*, *llsG*, *llsH*, *llsX*, *lpeA*, *lpA1*, *lspA*, *oatA*, *pdgA*, *prsA2* (17)

**Table 4 microorganisms-10-00512-t004:** Susceptibility testing results of the 14 *L. monocytogenes* isolates for antimicrobial agents with existing clinical breakpoints.

Antimicrobial Agent(s)	Susceptible		Intermediate		Resistant	
	no.	%	no.	%	no.	%
Penicillin	14	100	0	0	0	0
Ampicillin	14	100	0	0	0	0
Amoxicillin/clavulanic acid	8	57	6	43	0	0
Erythromycin	14	100	0	0	0	0
Clindamycin	0	0	0	0	14	100
Pirlimycin	0	0	0	0	14	100
Ciprofloxacin	14	100	0	0	0	0
Enrofloxacin	13	93	1	7	0	0
Marbofloxacin	12	86	2	14	0	0
Gentamicin	14	100	0	0	0	0
Streptomycin	14	100	0	0	0	0
Neomycin	14	100	0	0	0	0
Tetracycline	14	100	0	0	0	0
Doxycycline	3	21	11	79	0	0
Sulfamethoxazole/trimethoprim	14	100	0	0	0	0
Vancomycin	14	100	0	0	0	0

## Data Availability

The raw NGS data can be found for downloading at: https://www.ncbi.nlm.nih.gov/sra/PRJNA797842; accessed on 23 January 2022 (BioProject accession number PRJNA797842).

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
