# Peer review of "Distribution and Characteristics of Listeria spp. in Pigs and Pork Production Chains in Germany"

_microorganisms, 2022, doi:10.3390/microorganisms10030512_

Round 1

Reviewer 1 Report

The manuscript researched about Listeria monocytogenes, presented a whole genome sequence of the strains.

I believe that the results are important for the publication, but I suggest some modifications.

In line 93: "The isolates were recovered from..."

In item 3.1, I suggest to describe how many strains were isolated from each slaughterhouse. would be interesting compare the dates which the samples were taken.

I suggest to present a phylogenetic tree of strains, and the resistance genes observed in the sequences.

Author Response

Point 1: In line 93: "The isolates were recovered from..."

Response 1: Thank you for this improvement, we have changed it accordingly in the text.

Point 2: In item 3.1, I suggest to describe how many strains were isolated from each slaughterhouse. would be interesting compare the dates which the samples were taken.

Response 2: Thank you for this valuable input to get more information across to the reader. We added the information how many strains came from each slaughterhouse as well as the information about the sampling date in the Tables 2 and 3 for each isolate, so the dates can be compared for each isolate. As a result, the dates of the environmental samples in Section 3.2 and Table 1 had to be renamed.

Point 3: I suggest to present a phylogenetic tree of strains, and the resistance genes observed in the sequences.

Response 3: Many thanks for this suggestion. Considering that trees based on hierarchical clustering are already part of the manuscript (see Figure 1, Figure 2 and Figure 3), we see no added value in additionally showing a phylogenetic tree. The information most relevant to our conclusions are the SNP distances, which we used as an indication of possible epidemiological links/transmission pathways. In the included hierarchical clustering based on SNP analysis, this information is presented in a much clearer form than would be the case with a phylogenetic tree. Therefore, we would prefer to stay with this form of presentation for easier readability. Detailed information on the resistance genes for each isolate is displayed in Tables 2 and 3.

Reviewer 2 Report

Dear Authors,

Please refer to my (positive) comments in the attached file. In my opinion, your manuscript deserves publication after few minor editing amendments.

Author Response

Point 1: L16: Suggest replacing ‘older’ with elderly

Response 1: Thank you for this suggestion, we replaced it.

Point 2: L22: ….as well as in the pork production……

Response 2: We changed this sentence as suggested.

Point 3: L29 and elsewhere: apathogenic or non-pathogenic as the authors mention in later text points? Both terms are clear, however, I deem non-pathogenic is preferable linguistically.

Response 3: Following this suggestion, the word ‘apathogenic’ was changed to ‘non-pathogenic’.

Point 4: L95: …..later referred to as slaughterhouses A and B.

Response 4: Thank you for this comment, we corrected it.

Point 5: L97: environment isolated? The verb is missing: were isolated, or were obtained.

Response 5: Many thanks for spotting this mistake, it has now been corrected.

Point 6: L97-98: Please check and rewrite this sentence for clarity: …..sampling was performed, or samplings were performed, or better In slaughterhouse A samples were collected from….. then delete ‘collected’ Also, what does ‘over winter 2018/2019’ mean? Did you collect the samples during the two sequent winter periods only? Please clarify.

Response 6: Thank you for pointing this out, we corrected the sentence accordingly and added additional information about the sampling dates, as also Reviewer 1 asked for a specification of the sampling dates. We added more details about the sampling dates of each isolate in Tables 2 and 3 for making it possible for the reader to compare the isolates of each date with each other. We hope the information about the dates is clearer now.

Point 7: L104: Suggest replacing the comma with semicolon symbol after October 2019.

Response 7: Thank you, we implemented this suggestion.

Point 8: L106: …..isolates were stored in……

Response 8: Thank you, the sentence has been corrected.

Point 9: L128: Please check if any letters are missing before the digits in the accession number NC_003212.1.

Response 9: We checked it, the accession number is correct.

Point 10: L139: Why did you perform the phenotypic antimicrobial susceptibility testing for the L. monocytogenes isolates only? I deem it would be useful to test the other Listeria spp. also.

Response 10: This is a very good point that we have discussed also in our team. Our main point was that the few Listeria-specific clinical breakpoints available in the CLSI document M45 (3rd Edition) are valid exclusively for Listeria monocytogenes. This means that for isolates of other Listeria species, we cannot classify the MIC values into the categories susceptible, intermediate or resistant.

Point 11: L290: adaption, or adaptation?

Response 11: Thank you for spotting this, ‘adaptation’ is the right word here and has been correct in the text.

Round 2

Reviewer 1 Report

The authors  improved the manuscript as was suggested. I recommend the publication.